# Psychosocial problems, daily functioning and help-seeking behaviour of international migrant workers in the Netherlands: A qualitative study to inform the adaptation of a scalable stepped-care intervention

Rinske Roos[1,2] [ID], Anke B. Witteveen[1,2] [ID], Corrado Barbui[3], Richard Bryant[4] [ID], Zlata Dontsova[1], David McDaid[5] [ID], Josep Maria Haro[6,7], Kerry R. McGreevy[8,9], Roberto Mediavilla[8,9], Maria Melchior[10] [ID], Pablo Nicaise[11], A-La Park[12], Papoula Petri-Romão[13], Marianna Purgato[3] [ID], Aurélia Roversi[10], Annemieke van Straten[1,2], James Underhill[14] and Marit Sijbrandij[1,2] [ID]

[1]Department of Clinical, Neuro- and Developmental Psychology, WHO Collaborating Center for Research and Dissemination of Psychological Interventions, Vrije Universiteit Amsterdam, Amsterdam, Netherlands; [2]Amsterdam Public Health Research Institute, Amsterdam, Netherlands; [3]Department of Neurosciences, Biomedicine and Movement Sciences, WHO Collaborating Centre for Research and Training in Mental Health and Service Evaluation, University of Verona, Verona, Italy; [4]School of Psychology, University of New South Wales, Sydney, Australia; [5]Policy and Evaluation Centre, Department of Health Policy, London School of Economics and Political Science, London, UK; [6]Parc Sanitari Sant Joan de Deu, Institut de Recerca Sant Joan de Déu (IRSJD), Sant Boi de Llobregat, Barcelona, Spain; [7]Centro de Investigación Biomédica en Red de Salud Mental (CIBERSAM), Sant Boi de Llobregat, Barcelona, Spain; [8]Department of Psychiatry, Universidad Autónoma de Madrid, Madrid, Spain; [9]Centro de Investigación Biomédica en Red de Salud Mental (CIBERSAM), Instituto de Salud Carlos III, Madrid, Spain; [10]INSERM, Institut Pierre Louis d'Épidémiologie et de Santé Publique (IPLESP), Equipe de Recherche en Epidémiologie Sociale, Santé Mentale et Addictions (ESSMA), Sorbonne Université, Paris, France; [11]Institute of Health and Society (IRSS), UCLouvain, Brussels, Belgium; [12]Care Policy and Evaluation Centre, Department of Health Policy, London School of Economics and Political Science, London, UK; [13]Leibniz Institute for Resilience Research, Mainz, Germany and [14]Independent Research Consultant, Brighton, UK

## Research Article

**Keywords:**
adaptation; distress; global mental health; migrant workers; brief interventions

**Corresponding author:**
Rinske Roos;
Email: r.roos@vu.nl

## Abstract

International migrant workers (IMWs) may face insecure work and housing, limited access to healthcare and increased risk of psychological problems. Two scalable, evidence-based interventions to support individuals experiencing psychological distress are Doing What Matters in Times of Stress (DWM) and Problem Management Plus (PM+). This study aimed to explore IMWs' problems, daily functioning and help-seeking behaviour, to inform cultural adaptation of the DWM/PM+ stepped-care intervention in the Netherlands. Following the Design, Implementation, Monitoring, and Evaluation (DIME) model, we conducted various qualitative interviews and a focus group discussion with IMWs (n = 30) and professionals (n = 18). Data were analysed thematically, and findings informed adaptations. Participants described problems related to work, housing, administration, finances, healthcare access and the COVID-19 pandemic. Daily routines focused on practical needs. Help-seeking was hindered by stigma, fear of job loss, low trust and reliance on informal or cross-border healthcare. Based on these results, the intervention was adapted to the needs of Polish IMWs in the Netherlands, regarding content and examples, which were tailored to their context; the intervention was offered remotely and collaboration with employers was avoided. These findings highlight the structural vulnerabilities of IMWs and demonstrate how qualitative insights can guide the cultural adaptation of a psychological intervention.

## Impact statement

International migrant workers (IMWs) are essential to the global labour force, yet many face precarious working and living conditions alongside migration-related stressors. These challenges contribute to physical and mental health problems, while limiting their ability to seek support. Despite high levels of psychological distress, IMWs often struggle to access care due to language barriers, work schedules, restricted access to healthcare systems and limited awareness of available services.

This study explored IMWs' daily problems and help-seeking behaviour in the Netherlands, highlighting job instability, housing precarity, administrative barriers and dependency on employment agencies. These systemic obstacles not only increase mental health risks but also

shape how and when IMWs seek support. Findings informed the cultural adaptation of the World Health Organisation's (WHO) scalable interventions Doing What Matters in Times of Stress (DWM) and Problem Management Plus (PM+), for individuals experiencing psychological distress. Adaptations focused on enhancing feasibility and accessibility, including tailoring intervention content and modifying PM+ for remote delivery to better fit IMWs' circumstances. Although this study focused on IMWs in the Netherlands, the results may inform future adaptations of scalable mental health interventions for other migrant communities facing similar barriers.

These insights are relevant for mental health practitioners, policymakers and organisations working with migrant communities. They not only demonstrate how data on IMWs' lived experiences can inform the cultural adaptation of evidence-based interventions but also offer a comprehensive understanding of the structural barriers shaping their mental health risks and access to care. More broadly, this study underscores the importance of culturally responsive mental health interventions that take into account the lived realities of migrant workers, including the psychological impact of precarious work, housing instability and administrative barriers. By embedding such approaches into mental health policies and services, support systems can become more sustainable and inclusive for migrant populations worldwide.

## Introduction

International migrant workers (IMWs) are migrants of working age (≥15 years) in the labour force in their country of residence (ILO, 2021). In the Netherlands, this group includes both EU and non-EU workers. EU workers can work and live in the Netherlands without a visa or work permit, whereas non-EU workers typically need a visa and are often employed in high-skilled jobs (Immigration and Naturalisation Service, IND, n.d.; European Union, 2025). In 2020, almost one million jobs were held by foreign-born workers, over half from EU countries, particularly Poland (almost 40%) (CBS, 2022). Regardless of origin, all IMWs must obtain a citizen service number (BSN) by registering with the municipality and have Dutch health insurance, although these administrative steps are often arranged by employers (RVO NEA, n.d.-a, n.d.-b).

IMWs are overrepresented in so-called "3-D jobs": dirty, dangerous and demeaning (Moyce and Schenker, 2018; Norredam and Agyemang, 2019). Compared to non-migrants, they often work longer hours for less pay in unsafe conditions (Moyce and Schenker, 2018), experience more workplace accidents (Hargreaves et al., 2019) and labour exploitation (Mucci et al., 2020; Boufkhed et al., 2024), and are often overqualified for their jobs (Herold et al., 2024). Precarious living and working conditions increase their risk of physical and mental health problems (Mucci et al., 2020; Hasan et al., 2021; Ornek et al., 2022; Caxaj et al., 2024).

The COVID-19 pandemic further highlighted IMWs' vulnerability. Many were essential workers unable to work remotely, increasing their risk of infection (Fasani and Mazza, 2020), as illustrated by virus outbreaks in Dutch meat plants (Berntsen et al., 2022). Border closures and other measures further complicated their situation (Berntsen and Skowronek, 2021). Mental health also deteriorated more among vulnerable population groups, including young people, women and those with pre-existing health or socioeconomic issues (Pierce et al., 2021; Witteveen et al., 2022). A narrative review by Giorgi et al. (2020) identified IMWs as one of the most vulnerable occupational groups during the pandemic, due to a "double impact" of COVID-19-related stressors and adverse employment environment, including job insecurity, social isolation, labour rights exploitations and uncertainty about the future. These aspects interact with pandemic-related mental health problems, including anxiety, depression and PTSD, which are more likely to affect IMWs.

IMWs frequently report physical (e.g., lower back pain, respiratory infections) and mental health problems (e.g., anxiety, depression, post-traumatic stress, substance abuse) (Mucci et al., 2020; Hasan et al., 2021; Urrego-Parra et al., 2022). A systematic review of European studies found that unfavourable working conditions are associated with poor mental health, regardless of migrant status,

cultural origin or host country (Herold et al., 2024). Besides structural and occupational risks, mental health problems may be worsened by acculturative stress, the psychological strain of adapting to the host country's socio-cultural environment (Berry et al., 1987), which can be individual, social and work-related (Liem et al., 2021b). Depression among IMWs has been linked closely to socio-economic disadvantage, lack of family support and poor self-related health (Wang et al., 2019). Additionally, feelings of defeat, reflecting low social status and a sense of failure or loss in social struggle, are common among IMWs and strongly associated with anxiety and depression (Liu et al., 2023).

Despite these problems, IMWs underutilise health services compared to non-migrants (Pega et al., 2021). Barriers include limited awareness of services, cultural and language barriers, administrative complexities, financial constraints and work-related obstacles (Thomson et al., 2015; Czapka and Sagbakken, 2016; Hennebry et al., 2016; Ang et al., 2017). Previous studies have called for interventions to improve IMWs' wellbeing (Hargreaves et al., 2019; Giorgi et al., 2020; Mucci et al., 2020; Urrego-Parra et al., 2022). To our knowledge, there are no psychological interventions tailored to the needs of IMWs, despite growing calls for accessible, contextually appropriate mental health support for this population.

The World Health Organisation (WHO) has developed various scalable psychological interventions, originally designed for low-resource and humanitarian settings, including Doing What Matters in Times of Stress (DWM) and Problem Management Plus (PM+). DWM is a guided self-help web application based on acceptance and commitment therapy (ACT), with five weekly modules adapted from the Self-Help Plus (SH+) book (Epping-Jordan et al., 2016). PM+ is a transdiagnostic psychological intervention based on cognitive behavioural therapy, consisting of five weekly, 90-min in-person individual sessions (Dawson et al., 2015). Both SH+ and PM+ have demonstrated (cost-)effectiveness in reducing psychological distress (Tol et al., 2020; Acarturk et al., 2022; Park et al., 2022; Schäfer et al., 2023). Combined into a stepped-care intervention, DWM/PM+ has demonstrated effectiveness among healthcare workers in Spain (Mediavilla et al., 2023) and migrants in Italy (Purgato et al., 2025).

Before assessing an intervention's effectiveness, adaptation to the cultural and contextual needs of the target population is crucial, as adapted interventions are more effective than non-adapted ones (Chowdhary et al., 2014). This involves systematically modifying treatment protocols to ensure alignment with the target population's language, culture and context (Bernal et al., 2009). Such adaptations can enhance the intervention's acceptability, comprehensibility, relevance and completeness, while preserving its core elements (Bernal et al., 2009). Other contextual factors, such as the

COVID-19 pandemic, should also be considered (Akhtar et al., 2021).

In light of this, the aim of this formative study was to explore the perspectives of both IMWs and professional stakeholders regarding the problems, daily functioning, and help-seeking behaviour of IMWs, to inform the cultural adaptation of the DWM/PM+ stepped-care intervention for IMWs in the Netherlands. The study focused on IMWs in low-skilled work. Research questions were (a) *What problems do IMWs in the Netherlands face?* and (b) *How can these insights inform the cultural adaptation of the DWM/PM+ stepped-care intervention?*

## Methods

This study is part of the RESPOND (Improving the PREparedness of Health Systems to Reduce Mental Health and Psychosocial Concerns resulting from the COVID-19 PaNDemic) project. It followed an exploratory, participatory design, involving participants in identifying IMW problems and needs and informing trial implementation (Roos et al., 2023).

### Research design

Following the DIME (Design, Implementation, Monitoring and Evaluation) framework (Applied Mental Health Research Group, 2013), three rounds of qualitative interviews were conducted with IMWs:

1. **Round I – Free Listing (FL) interviews:** Participants listed what came to mind concerning (a) daily life problems (general and COVID-19-related) and (b) daily functioning activities (self-care, family care and community support). Next, they elaborated on each topic and described characteristics of IMWs "doing well" versus "not doing well."
2. **Round II – Key Informant (KI) interviews:** Semi-structured interviews explored key problem themes identified in FL interviews.

3. **Round III – Focus Group Discussion (FGD):** Participants reviewed and ranked FL daily functioning activities, discussed help-seeking behaviour and gave feedback on DWM/PM+ (content, suitability, delivery, perceived barriers).

In parallel, semi-structured interviews with professional stakeholders explored IMWs' daily problems, mental health and service accessibility. They also addressed the DWM/PM+ intervention, including its suitability, potential health system integration, target population and recruitment strategies. Figure 1 provides an overview of the data collection process.

### Sample and recruitment

Inclusion criteria for IMWs were broad, allowing anyone self-identifying as an IMW living in the Netherlands, regardless of country of origin, duration of stay, work status or sector. Professionals needed current or recent experience working with IMWs. No exclusion criteria were applied. Recruitment followed predefined target ranges (i.e., 20–24 FL interviews, 10–15 KI IMWs, 15–20 KI professionals, 1–2 FGDs with 5–8 participants each) and concluded due to time constraints.

Although recruitment relied on convenience and snowball sampling, we aimed for maximum variation sampling to ensure diverse perspectives (Patton, 1990). IMWs were primarily recruited through Facebook groups (e.g., "Polish people in the Netherlands"). FL participants were asked to refer others for KI interviews (Round II), although only one IMW was recruited through this method. Professionals were approached via municipalities, non-governmental organisations (NGOs) and healthcare services. Variation was sought in professional roles, and, for IMWs, in gender, age and country of origin.

### Procedure and setting

All interviews took place between February and May 2021, before COVID-19 vaccinations were widely available in the Netherlands. Interviews were conducted via telephone or Skype for Business, in Dutch, English, Polish, Ukrainian or Russian, by trained multilingual

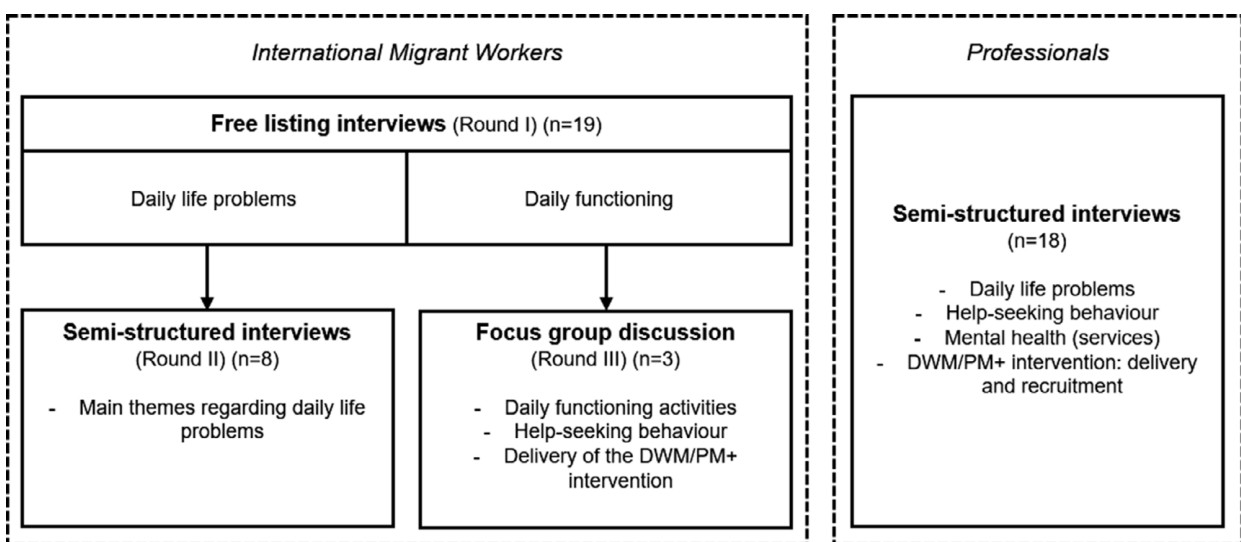

**Figure 1.** Overview of the qualitative data collection process, including the types of interviews and main topics discussed. Findings from Free Listing Interviews (Round I) with international migrant workers informed the semi-structured interviews (Round II) and focus group discussion (Round III). Interviews with professionals consisted of semi-structured interviews only

research assistants (Polish and Latvian) and the first author (Dutch). Interviews were audio recorded using a separate device for confidentiality. Semi-structured interviews lasted 45–90 min, the FGD 90 min. IMWs were offered €10 (FL) or a €20 gift-voucher (KI/FGD); professionals were not compensated.

### Analysis

FL and FGD responses were documented in reporting sheets. Non-Dutch/English interviews were translated into English by research assistants. All data analysis was conducted solely by the first author (RR), who has a background in psychology (bachelor's) and global health (research master's focused on mixed-methods research, cross-cultural health and health inequalities). Although she does not share the culture or language of the target group, her contextual understanding was developed through direct engagement with IMWs and stakeholders during this study, for which she conducted most data collection (all interviews with professionals and half of the IMW interviews), as well as during the subsequent trial (Roos et al., 2023), for both of which she served as main researcher. No inter-coder reliability checks or investigator triangulation were performed.

FL problems were grouped thematically in Excel. All KI interviews were manually transcribed and analysed using inductive thematic analysis (Braun and Clarke, 2006) in Atlas.ti (version 23.3.4), by coding meaning-bearing sections without a predefined codebook. Codes were grouped into broader themes based on recurring patterns. Throughout the coding and analysis, special attention was paid to closely grounding all codes and themes in the explicit statements and perspectives of participants, in order to minimise interpretive bias. Saturation was evaluated during the coding process and considered reached, as no new themes or substantial new information were identified in the final interviews. Findings from the FL, KI and FGD interviews informed adaptations to the DWM/PM+ intervention. Adaptations were categorised using Bernal et al.'s (2009) eight domains *(language, persons, metaphors, content, concepts, goals, methods and context)*.

### Ethics

The study was approved by the Scientific and Ethical Review Board of the Vrije Universiteit Amsterdam (VCWE-2021-006). Participants received a Dutch, English or Polish information letter and provided written (digital) consent before participation.

### Results

#### Participants

A total of 48 individuals participated: 30 IMWs and 18 professionals. Table 1 summarises IMWs' sociodemographic characteristics. Professionals worked in healthcare (n = 6), NGOs (n = 8) and governmental organisations (n = 4), mostly with Polish migrant workers.

#### Profile of IMWs

KI professionals described how most IMWs come from Poland, with growing numbers from Romania, Bulgaria and Hungary, especially for seasonal labour. Many are in their thirties and forties, although more young adults now migrate directly after high school. IMWs were described as generally lower-educated, often not

**Table 1.** Sociodemographic characteristics of international migrant workers in the free listing (FL) interviews (Round I), key informant (KI) interviews (Round II), and focus group discussion (FGD) (Round III)

| Variable | FL (n = 19) | KI (n = 8) | FGD (n = 3) | Total (N = 30) |
|---|---|---|---|---|
| Gender, n (%) | | | | |
| Male | 9 (47) | 4 (50) | 0 (0) | 13 (43) |
| Female | 10 (53) | 4 (50) | 3 (100) | 17 (57) |
| Age, M (SD) | 33.9 (9.4) | 34 (5.8) | 28.7 (8.3) | 33.4 (8.4) |
| Country of origin, n (%) | | | | |
| Greece | 1 (5) | 0 (0) | 0 (0) | 1 (3) |
| Hungary | 0 (0) | 1 (13) | 0 (0) | 1 (3) |
| Peru[a] | 1 (5) | 0 (0) | 0 (0) | 1 (3) |
| Poland | 10 (53) | 7 (88) | 3 (100) | 20 (67) |
| Romania | 2 (11) | 0 (0) | 0 (0) | 2 (7) |
| Slovenia | 1 (5) | 0 (0) | 0 (0) | 1 (3) |
| Spain | 1 (5) | 0 (0) | 0 (0) | 1 (3) |
| Ukraine[a] | 3 (16) | 0 (0) | 0 (0) | 3 (10) |
| Total time in the Netherlands (years), M (SD) | 5.0 (5.0) | 6.0 (3.8) | 7.5 (8.8) | 5.5 (5.0) |
| Highest completed education level, n (%) | | | | |
| High school or equivalent | 4 (21) | 2 (25) | 0 (0) | 6 (20) |
| Vocational/Technical/ Trade | 3 (16) | 0 (0) | 2 (67) | 5 (17) |
| University level (Bachelor's/Applied Sciences) | 4 (21) | 4 (50) | 1 (33) | 9 (30) |
| Graduate (Master's/ Doctoral) | 8 (42) | 2 (25) | 0 (0) | 10 (33) |
| Work status, n (%) | | | | |
| Unemployed | 3 (16) | 3 (38) | 1 (33) | 7 (23) |
| Part-time | 2 (11) | 0 (0) | 1 (33) | 3 (10) |
| Full-time | 11 (58) | 4 (50) | 1 (33) | 16 (53) |
| Self-employed | 1 (5) | 0 (0) | 0 (0) | 1 (3) |
| On leave (sick, maternity) | 0 (0) | 1 (13) | 0 (0) | 1 (3) |
| Highly variable hours | 1 (5) | 0 (0) | 0 (0) | 1 (3) |
| Missing | 1 (5) | 0 (0) | 0 (0) | 1 (3) |
| Work sector, n (%) | | | | |
| Agriculture | 0 (0) | 2 (25) | 0 (0) | 2 (7) |
| Cleaning | 1 (5) | 0 (0) | 0 (0) | 1 (3) |
| Construction | 1 (5) | 1 (13) | 0 (0) | 2 (7) |
| Food sector | 2 (11) | 0 (0) | 0 (0) | 2 (7) |
| Human resources management | 1 (5) | 2 (25) | 1 (33) | 4 (13) |
| Healthcare | 1 (5) | 0 (0) | 0 (0) | 1 (3) |
| Hospitality | 0 (0) | 1 (13) | 0 (0) | 1 (3) |

*(Continued)*

**Table 1.** (*Continued*)

| Variable | FL (n = 19) | KI (n = 8) | FGD (n = 3) | Total (N = 30) |
|---|---|---|---|---|
| IT | 1 (5) | 0 (0) | 0 (0) | 1 (3) |
| Logistics | 6 (32) | 1 (13) | 1 (33) | 8 (27) |
| Metal industry | 1 (5) | 0 (0) | 0 (0) | 1 (3) |
| Production | 2 (11) | 0 (0) | 1 (33) | 3 (10) |
| Research | 1 (5) | 0 (0) | 0 (0) | 1 (3) |
| Retail | 2 (11) | 1 (13) | 0 (0) | 3 (10) |
| Living situation, n (%) | | | | |
| Alone | 4 (21) | 2 (25) | 0 (0) | 6 (20) |
| With partner | 7 (37) | 0 (0) | 2 (67) | 9 (30) |
| With partner + child(ren) | 1 (5) | 3 (38) | 1 (33) | 5 (17) |
| With relatives | 1 (5) | 1 (13) | 0 (0) | 2 (7) |
| With friends | 2 (11) | 0 (0) | 0 (0) | 2 (7) |
| With colleagues | 0 (0) | 1 (13) | 0 (0) | 1 (3) |
| With others | 4 (21) | 1 (13) | 0 (0) | 5 (17) |

[a]Participants from non-EU countries, who are not covered by the EU principle of free movement and may therefore face more restrictive conditions regarding work and residence in the Netherlands.

speaking a second language (i.e., English), from rural areas, and in some cases facing literacy challenges. Many reportedly arrive with pre-existing family, financial or legal issues and lack skills or confidence to navigate systems abroad. Combined with fear of job loss, this makes newer migrants particularly vulnerable to exploitation.

Many IMWs reportedly arrive alone or with a partner, primarily to earn money, but also to prove themselves or boost their self-esteem. They were said to receive poor or misleading information before migrating, often focused on wages rather than living conditions. Some aim to save for specific goals (e.g., loans, home down-payment), while others face pressure to financially support family back home. Although most plan to stay temporarily, they keep extending their time with "one more year." Together with the absence of integration requirements for EU citizens and access to a so-called "Polish parallel world" in the Netherlands, this was said to foster a state of "long-term temporariness," where learning Dutch is continually postponed.

### Daily life problems

During FL interviews (Round I), IMWs identified 154 daily-life problems, most of which were related to COVID-19 (102 COVID-19 related, 52 general), which were coded into 33 categories (20 COVID-19 related, 13 general; see Figure 2). Based on these categories, KI interviews with IMWs (Round II) explored four COVID-19-related topics: "lockdown measures," "international travel restrictions," "unemployment and financial problems" and "work and housing." These were selected shortly after the FL interviews, based on prominent problems and their urgency and relevance in early 2021, when pandemic-related stressors were especially pressing. KI interviews with professionals explored broader issues affecting EU workers, as non-EU workers face distinct challenges.

The results of KI interviews with both IMWs and professionals, are organised below into eight problem themes. The first six reflect structural and practical challenges that IMWs routinely face, and often intensified during the pandemic. Where relevant, we distinguish between general and COVID-specific issues within a theme. The last two themes focus specifically on challenges introduced or exacerbated by COVID-19 lockdown measures. Table 2 provides an overview of the main findings per problem theme, with illustrative quotes.

1) *Work and unemployment:* Participants described a range of problems faced by IMWs working via private employment agencies, which often also arrange housing, health insurance and transport. While this arrangement may seem attractive, it was criticised for creating multi-layered dependency. IMWs were said to often work on temporary contracts with limited rights (e.g., no sick pay/notice period), in unskilled, physically demanding shift jobs with heavy workloads and expected over-time. Some were reportedly unaware they would work for an agency until arrival and were misled about English language requirements, which could lead to dismissal within days for not meeting expectations.

Cultural and language barriers reportedly cause misunderstandings with employers. Some IMWs reportedly work without taking holidays, either fearing dismissal or unaware vacation days were paid weekly. Discrimination was also mentioned, with locals prioritised for better conditions. To secure jobs, competition between IMWs was reportedly common. Agencies were said to exploit workers through contract changes, illegal health insurance cancellations, dismissal threats following complaints and repeated termination before qualifying for more stable contracts, keeping IMWs in cycles of unstable employment.

COVID-19 exacerbated employment-related vulnerabilities. Participants highlighted unsafe working conditions, including inadequate protective measures and agency pressure to avoid testing or quarantine, forcing IMWs to choose between working without proper protection or being dismissed. COVID-positive IMWs were sometimes pressured to keep working, while others were evicted from employer-provided accommodation to prevent other agency employees from being quarantined. Fear of losing their accommodation prevented some from reporting illness, fostering distrust among IMWs.

Financial insecurity reportedly also worsened during COVID-19, particularly for those on temporary contracts, without sick pay or access to social benefits. Participants described reduced hours, delayed wages and job losses. Weekly wage payments reportedly left many without savings, making basic needs difficult to afford when unemployed. Language barriers, lack of formal qualifications and the closure of sectors further limited job opportunities, particularly for older IMWs and those without a driver's license. As a result, many reportedly felt compelled to accept poor conditions.

2) *Housing:* Housing was a major concern, particularly employer-provided accommodation. While living conditions vary (e.g., holiday parks, IMW-hotels, urban neighbourhoods), participants described problems such as overcrowding, shared bed-rooms, poor maintenance, limited facilities and a high housemate turnover, causing conflicts and insecurity. They highlighted how IMWs' dependence on staffing agencies for both work and housing made them vulnerable: job loss could result in eviction and, at times, immediate homelessness. Financial barriers (i.e., low wages, high rents), waiting lists, language barriers, lack of transport and housing shortages were said to

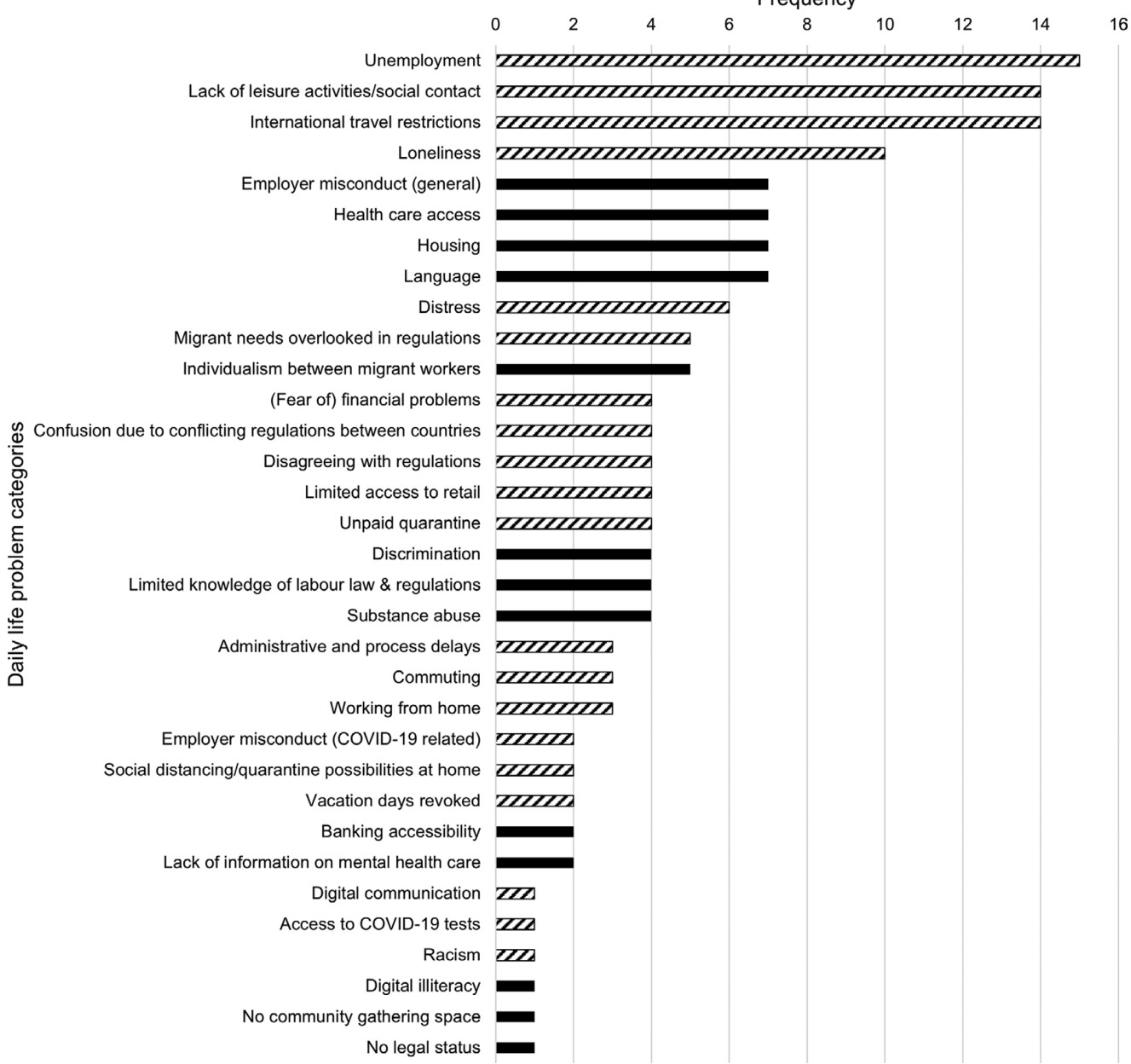

**Figure 2.** Frequency of daily life problems identified by international migrant workers during the free listing interviews (Round I): problems thematically grouped as either COVID-19-related (striped bars) or general, non-COVID-related (solid bars)

prevent access to independent housing, forcing many into substandard accommodations.

 During COVID-19, these housing issues reportedly intensified. Shared accommodations made social distancing and quarantining difficult. As job losses increased, participants noted a rise in homelessness. While some returned home, others could not afford to, leaving them stranded in the Netherlands.

3) *Administration:* Participants highlighted administrative challenges, with many IMWs arriving unprepared and unfamiliar with Dutch bureaucracy. Not registering with the municipality upon arrival was considered a key barrier, preventing access to legal employment, a Dutch bank account and social benefits. Some noted limited availability of information in IMWs' native languages or difficulty finding or interpreting it. Employment agencies reportedly provide little guidance, increasing the risk of bureaucratic complications.

 IMWs were said to struggle with the Netherlands' emphasis on individual responsibility (e.g., tax filings, welfare applications), which often requires navigating unfamiliar online tools. Limited digital literacy, fear of making mistakes and distrust of government institutions, linked by some to IMWs' countries of origin, discourages them from handling paperwork independently. Instead, they reportedly rely on peers or expensive private accountants, whose advice was considered unreliable and led to mistakes, mismanaged subsidies or debt. Even after years in the Netherlands, many IMWs reportedly remain unaware of their rights.

**Table 2.** Key themes, main findings and example quotes on daily life problems of international migrant workers (IMWs), based on semi-structured interviews (Round II) with IMWs and semi-structured interviews professionals

| Theme | Main findings | Example quote |
|---|---|---|
| 1) Work and unemployment | - IMWs are highly dependent on employment agencies, which control work, housing and insurance;<br>- Precarious contracts, lack of sick pay and frequent dismissals create ongoing job insecurity;<br>- Exploitative practices and discrimination are common; IMWs have limited options to improve their situation;<br>- COVID–19 made conditions worse: more job losses, unsafe workplaces and pressure to work while sick. | - *"The problem these people face when they come to work in the Netherlands is multiple dependencies on someone who recruits them for work, and then ties their housing to their wages, and their transportation to their wages as well. (…) And that, of course, puts them in a position of dependence, where they might not prioritize their rights as much as they would in a different situation. (…) You are vulnerable, essentially, when it comes to safe, healthy, and fair work."* (Profi13, governmental organisation)<br>- *"I think they are treated differently from Dutch workers. I'm talking about working conditions, where employers very easily assume that they will work overtime or that, instead of working 8 h, they will work 10. That is very common."* (Profi15, health care)<br>- *"They simply accept worse working conditions, that is the boss pays them less. And they know it's not okay. But it's better to be paid less than not at all."* (IMW1, male, 36 yrs, Poland)<br>- *"They work as if for two or three people because they are so determined to keep their job and not lose it. (…) If they work too hard, they get physical problems. If they can't work, they get psychological problems."* (Profi18, health care)<br>- *"If you say to your labour provider, 'I don't want to get in this van because you haven't arranged proper COVID–19 measures,' the provider will say: 'Well, fine, pack your bags, we have someone else for your spot.'"* (Profi13, governmental organisation)<br>- *"Most of us are just here to make money. (…) They sacrifice a lot of things, just so their families in Romania and in Poland live a better life. (…) So even if they have symptoms of the disease, they go to work, hide it so as not to be dismissed and stay in quarantine, because quarantine means either unpaid leave or 70% paid leave. Generally, it is never 100% of your salary."* (IMW1, male, 36 yrs, Poland) |
| 2) Housing | - Employer-arranged housing limits IMWs' independence and security; job loss may lead to immediate eviction and homelessness;<br>- Living conditions are frequently poor: overcrowding, inadequate facilities and high turnover are common;<br>- IMWs face high costs, lack of affordable alternatives and fear retaliation for reporting poor conditions;<br>- COVID–19 intensified housing insecurity, making social distancing impossible and increasing evictions and homelessness. | - *"I've also lived in housing that was nice, clean. But I've also been in other places that were completely different, real chaos. I lived in housing meant for 5 or 6 people, but there were 15 of us. That was a disaster, of course."* (IMW8, male, 37 yrs, Poland)<br>- *"We lived with 14 people in that house, but not always the same ones. The Polish guys just come and go, come and go. (…) Sometimes I don't know their names."* (IMW5, female, 46 yrs, Hungary)<br>- *"[The law] states that everyone must have 12 square meters at their disposal. Well, let's just say that this requirement is not always taken into account."* (Profi13, governmental organisation)<br>- *"You have these 'Polish-hotels'. And there, you often have between 40 and 200 Polish men, either in large halls or in small rooms with two or three people."* (Profi11, health care)<br>- *"You also often see (…) that people rent a small room from someone. And they are not registered because of housing benefits; the homeowner would lose their allowance if they officially registered someone at the address."* (Profi1, NGO)<br>- *"Because work is often tied to housing, or rather, housing is tied to work. It means that the moment they lose their job, they also lose their home. And that happens very aggressively. It can mean that they're expected to leave their house the very next day. That's where I see a huge increase in stress."* (Profi9, NGO)<br>- *"I also saw (…) many people who fell into homelessness or temporary homelessness, because at the time [during COVID–19] when the agency was losing its client, it did not need as many people as it had before. It was terminating employment contracts for these people, but (…) the employment contract was the agreement to rent beds in the room."* (IMW1, male, 36 yrs, Poland) |
| 3) Administration | - IMWs often arrive unprepared for Dutch bureaucracy, with little support from agencies;<br>- Not registering with the municipality blocks access to work, housing, banking and benefits;<br>- Language barriers, limited digital skills and distrust of authorities complicate paperwork;<br>- Many rely on peers or paid intermediaries for help, increasing risk of errors and debt. | - *"Some people end up staying here for 10 or even 15 years and still have never fully understood how to apply for unemployment benefits, how to register with the municipality, or how to sort out all the basic aspects of life in the Netherlands."* (Profi11, health care)<br>- *"On our website, the information is available in three languages, but not in eight. So for some countries, for certain nationalities, there is much less information available."* (Profi8, governmental organisation)<br>- *"They are in housing with other Romanian people or maybe also, Polish or other migrant workers who don't have any more information either."* (Profi17, NGO)<br>- *"They go straight to an accountant who speaks their language and takes care of everything for them. But if that accountant turns out to be unreliable, or if they can no longer afford to pay for one, then they are left stranded."* (Profi2, NGO) |

*(Continued)*

**Table 2.** *(Continued)*

| Theme | Main findings | Example quote |
|---|---|---|
| 4) Physical health and access barriers | - IMWs face significant barriers to healthcare access: registration, insurance, language and cost;<br>- Fear of job loss and lack of information leads to delays in seeking care, self-medication or going without treatment;<br>- Mistrust of Dutch healthcare (e.g., focus on self-care, limited access to specialists) drives some to seek care abroad or via informal networks. | - "The system is not designed for migrant workers, so they don't understand it. They don't know that they need to go to a GP first. They think, 'I have problems with my lungs, I need to see a lung specialist,' because that's what you do in Romania" (Prof17, NGO)<br>- "Less trust in the healthcare system, for sure. There's the issue with antibiotics and paracetamol. People always take you seriously. Even for the smallest complaint, they will prescribe something. And in the Netherlands it's more like, 'Let's wait and see, just take some paracetamol.' And then they really think, 'No way...'" (Prof2, NGO)<br>- "I even know people who get in their car and drive back to their home country because here, they feel unseen, misunderstood, and not taken seriously. That's the issue, not being taken seriously. Not to move back completely, but literally just to see a GP, or to go to the hospital" (Prof8, governmental organisation)<br>- "Polish people bring antibiotics from Poland illegally to distribute between them. If you look at the Facebook group (…) you'll often see people asking: 'Does anyone have this or that type of antibiotics? Because if I go to the GP, of course, I won't get them." (Prof16, health care)<br>- "There is also the issue of drug use in the workplace. To work hard, younger people use a lot of speed and other drugs, that's well known. Among slightly older people, I think around 30, I mostly see marijuana use. They experience a lot of pain, especially in their backs, from the heavy loads they have to lift every day. These people also struggle to see a GP. All the GPs in our region are full, so they try everything they can to get an appointment, including through health insurance, but they don't succeed. So yes, they just look for ways to relieve the pain. And marijuana, apparently, is an effective remedy for that." (Prof14, NGO) |
| 5) Mental health and access barriers | - IMWs face high rates of stress, anxiety, depression and burnout, driven by harsh work conditions, financial insecurity and isolation;<br>- Stigma and fear of job loss prevent open discussion of mental health or timely help-seeking;<br>- Barriers to care include long waiting lists, lack of language-matched providers, high costs and difficulty navigating the system;<br>- Many rely on substance use or informal networks to cope, rather than formal support. | - "There are not enough psychologists or centres that can provide help. And even if there are, the waiting times are as long as 7, 8 or 10 weeks. Since these people are already at the end of their endurance, they rather decide to return to their country, seek help there." (IMW1, male, 36 yrs., Poland)<br>- "They experience stress very physically. They often come to me and say, 'I have high blood pressure, I feel tension in my chest or stomach.' (…) They are stressed about being late, people not understanding them, not being able to express themselves." (Prof14, NGO)<br>- "People don't want to admit they have problems so quickly. I think they tend to believe for too long that they can solve it themselves. And secondly, I think it is also a bit a matter of shame. That if I say, 'I'm going to a psychologist or psychiatrist,' it means I'm not completely okay. So you're immediately seen as crazy. It's not just that you have problems, like depression or something. But that you've completely lost it. And people don't want others to think that about them." (Prof1, NGO)<br>- "Sometimes, when it comes to addiction treatment, there's a translator available, and only then they can help. But sometimes, that translator is no longer available, or the funding runs out. And then people don't get help, because they are still expected to speak at least some Dutch, or English. But the thing is, they often don't speak Dutch. If you're working 5–6 days a week, 10 h a day. Then you go to a language class, and you don't learn anything." (Prof12, health care)<br>- "People are very stressed, and because they are in survival mode, they don't really know where to ask for help. They struggle to find someone they can trust for support. (…) The waiting lists are enormous. (…) The problems they face are often severe. Many have been through a lot in the past. (…) And then there's the lack of a network (…) because, of course, they are new here." (Prof2, NGO) |
| 6) Family | - Long work hours and lack of affordable childcare make it difficult for IMWs to care for their children and balance family responsibilities;<br>- Language barriers, unfamiliarity with Dutch child welfare policies and fear of child removal discourage help-seeking;<br>- Financial and emotional strain, family separation and limited support networks contribute to relationship stress and household tension;<br>- COVID–19 worsened these challenges, especially during school closures. | - "What we see within this project is that many more requests for help are related to family crises and divorce. Far more than in previous years. Because we've had this project for three years now. And before that, we saw a wide range of issues. But (…) many cases now involve relationship problems, people considering divorce, as well as stress and anxiety. So, actually, psychological challenges have increased." (Prof5, NGO)<br>- "The moment people have children, they are automatically afraid: 'If something happens, my child will be taken away in the Netherlands.' Of course, that never just happens. There are countless steps before it would ever come to that. But they are so afraid of it that it immediately becomes a taboo topic." (Prof10, health care) |

*(Continued)*

**Table 2.** *(Continued)*

| Theme | Main findings | Example quote |
|---|---|---|
| | | - *"Due to the lockdown, lessons were given digitally. This was especially difficult for children of migrant workers (…) Their parents work long hours. They definitely don't work from home because they are employed in production or distribution centres."* (Prof1, NGO)<br>- *"At school it's different; they go to school and can learn the language quickly. But with online lessons, parents can't help them. So I notice that parents worry a lot, [children] are deteriorating in terms of language. It's not just about sitting down and learning; their parents simply can't help them. So that gap keeps getting bigger and bigger."* (Prof6, NGO) |
| 7) Lockdown measures affecting daily life | - Lockdowns intensified social isolation, boredom and monotony in IMWs' daily lives;<br>- Closure of gathering spaces and language barriers limited opportunities for social contact and access to information;<br>- IMWs received little practical or employer support during quarantine;<br>- Media coverage of COVID–19 outbreaks increased distrust and stigma between IMWs and Dutch society. | - *"And you just sat in your rooms because there were restrictions everywhere. From my perspective, I can say this because we stayed in a very large hotel for such employees (…). There were sticky notes everywhere: 'You can't go out here, you can't go out here, you can't go out here.' So it was very, very tedious. Just work and back. It was tough. You had to sit locked up and wait for someone to explain what was happening."* (IMW2, male, 34 yrs., Poland)<br>- *"I think nothing changed, because if they have some colleagues, (…) some kind of friendship. And I think they still meet each other."* (IMW6, male, 32 yrs., Poland)<br>- *"A lot of people I have contact with reacted to work-related stress by going to the gym or to discos. (…) They were able to relieve stress that way. But when everything was closed, there was no place to let loose."* (IMW1, male, 36 yrs., Poland)<br>- *"Once a week, I would go to McDonald's or KFC, usually on Sundays. I liked it, watching people, seeing what they were doing, the families. But now I can't go. And that's really bad for me, because it was my only social activity."* (IMW5, female, 46 yrs., Hungary) |
| 8) Lockdown measures affecting international traveling | - Travel restrictions and quarantine rules disrupted IMWs' ability to visit family, causing distress and isolation;<br>- Financial, work-related and logistical barriers made travel difficult or unaffordable for many;<br>- IMWs relied on informal information networks to navigate changing regulations, sometimes leading to confusion. | - *"I think that migrant workers are afraid to travel because they can get infected, that they can be financially disadvantaged later, also in quarantine after returning to the country, to the Netherlands for example. I think that the financial aspects, above all, play a big role here."* (IMW4, female, 30 yrs., Poland)<br>- *"You stay in the Netherlands for as long until contract termination. Or until the moment of a higher force, if something bad happens at home, then you leave. Otherwise, you just stay in this country where there is work."* (IMW7, female, 28 yrs., Poland)<br>- *"My main problem, what really brings me down, is that I can't go [home]. If I go there, I have nowhere to stay because they [family/friends] would have to quarantine."* (IMW5, female, 46 yrs., Hungary) |

4) *Physical health and access barriers:* Participants highlighted multiple barriers to healthcare access. IMWs often register with Dutch general practitioners (GPs) only when health issues become serious, but many practices do not accept new patients, complicating access to urgent care. Difficulties navigating the system and language barriers further delay help-seeking and increase emergency visits. Uninsured IMWs reportedly face additional challenges, as some providers require upfront payment.

Participants also noted IMWs' mistrust of Dutch healthcare, particularly its focus on self-care. IMWs may perceive GPs as dismissive, typically recommending only basic treatments like paracetamol, which contrasts with the direct access to specialists and antibiotics in their home countries. Consequently, many reportedly seek care in their home countries when ill, traveling back and forth, while others self-medicate with medication sent from abroad or obtained through informal networks.

5) *Mental health and access barriers:* Participants reported significant mental health problems among IMWs, including stress, frustration, loneliness, somatic complaints, depression, anxiety, PTSD and burnout. These were linked to demanding work and living conditions, financial insecurity, social isolation and unresolved trauma. Substance use was frequently mentioned, with younger IMWs reportedly using drugs (e.g., speed, cannabis) facilitated by easy access and reduced social control, while older IMWs tend to turn to alcohol. Other coping strategies included music, prayer, church visits and isolation. Although socialising was seen as a source of support, participants noted that personal and work-related struggles are rarely discussed, fearing this information could be used against them. Instead, conversations with housemates or colleagues facing similar issues were said to reinforce distress, rather than alleviate it, offering no concrete solutions. Participants observed that these problems intensified during the COVID-19 pandemic, with increased stress, anxiety, depression, addiction (especially among men), eating disorders (among women) and suicidal concerns.

Access to mental healthcare was said to be limited even outside of the COVID-19 pandemic. Barriers mentioned included long waiting lists, few professionals speaking IMWs' language and difficulties navigating the system. Participants noted that Polish-speaking therapists are mostly concentrated in large cities, often not covered by insurance or too expensive. Additional barriers included inflexible work schedules and transportation, as IMWs reportedly avoid requesting time off, fearing it could jeopardise their job. Cultural expectations around therapy further complicate access: IMWs often expect directive care but encounter a system based on shared decision-making. Moreover, treatment durations were viewed as too short to address the complex interplay of psychosocial and practical problems IMWs face.

6) *Family:* Participants reported that IMWs with families in the Netherlands struggle to balance work and childcare. Long working hours limit parents' ability to support their children, while formal childcare is often underutilised due to high costs, limited awareness and cultural preferences for informal care. Without nearby relatives, young children are sometimes left unsupervised, have disrupted sleep or lack proper nutrition. Participants described shifting parent–child roles, as children integrate more quickly than their parents.

Cultural differences in parenting norms further complicate these challenges. Participants explained that IMWs often view Dutch child welfare policies as stricter than in their home countries, creating wariness towards social services. This was illustrated by social media warnings advising against bringing children to the Netherlands due to fears of child removal. Financial stress, long work hours and alcohol misuse were also said to contribute to relationship problems. Despite these difficulties, participants emphasised IMWs' strong commitment to their families, often at the expense of their own well-being.

During the COVID-19 pandemic, family-related difficulties worsened due to financial instability and prolonged confinement. School closures were considered particularly challenging for working (single) parents, as children lacked supervision and language barriers limited parental support with schoolwork. Due to childcare difficulties, some parents sent their children to family back home, raising concerns about disrupted education and social isolation.

7) *Lockdown measures affecting daily life:* Participants described IMWs' daily lives as monotonous, following a "work, eat, sleep, repeat" routine. Many prioritise saving money over social activities, form small groups with fellow nationals, and have limited interaction with Dutch society. Those working fewer hours reportedly experience boredom and frustration; drinking at home is a common way to pass time and cope with loneliness. Language barriers and a lack of accessible activities further contributed to social isolation. Participants emphasised the need for structured daytime activities (e.g., volunteering) to combat isolation and create meaningful routines.

COVID-19 intensified these problems. While some participants said lockdowns had minimal impact, as many IMWs tended to stay home already, most described how restrictions increased monotony, loneliness and boredom. Curfews, limits on visitors in employer-provided accommodation, and the closure of gyms, restaurants and language classes further reduced social contact. Younger IMWs reportedly adapted more easily to digital communication, while older generations found it insufficient. Without ways to unwind, some turned to substance use, while others used dating apps or organised small parties to cope.

Language barriers further complicated daily life during lockdowns. Participants noted that COVID-19 regulations were often only available in Dutch or English, making it difficult for IMWs to understand rules, schedule appointments, or access services. Store closures added practical difficulties, as deliveries required someone to stay home. Some participants wished for more employer support during quarantine (e.g., groceries). Finally, participants described increased distrust between IMWs and Dutch society, partly fuelled by media coverage of COVID-19 outbreaks in IMW accommodations.

8) *Lockdown measures affecting international traveling:* Participants described how travel restrictions disrupted IMWs' regular visits to their home countries. Many did not travel due to financial barriers, such as increased costs for tickets, COVID-19 tests, and unpaid quarantine or due to work obligations. Others feared sudden border closures, changing travel rules, or infection risks for themselves or (elderly) family members. Some avoided travel because quarantine would prolong absence from work.

Those who did travel reportedly relied on social media and Polish websites for advice, which sometimes conflicted with Dutch regulations. The inability to travel freely reportedly caused significant distress. Many missed important family moments (e.g., birthdays, funerals) or had responsibilities back home (e.g., businesses, sick parents). Most stayed connected via video calls and compensated by sending expensive gifts.

### Daily functioning

In the FL interviews (Round I), IMWs identified 101 daily functioning activities: 49 self-care, 22 family care, and 30 community-support activities (see Supplementary Tables S1–S3 for the full lists). When these lists were presented in the FGD (Round III), participants identified sports, alcohol use, walking, meeting friends and cooking as most common self-care activities, while meditation was specifically pointed out as something that IMWs do not practice. Regarding family care, IMWs primarily engage in daily conversations, send money or gifts, visit family back home and hide struggles. Within social networks, common support includes helping others find work, offering jobs, lending money, sharing meals and bringing traditional food or parcels from their home country. Practical help, such as offering rides or temporary housing, part of community support, was also frequently mentioned.

### Help-seeking behaviour

FGD participants (Round III) described how IMWs often avoid seeking help when facing difficulties, feeling resigned to problems being inevitable and beyond their control. They also noted a lack of autonomy and expectation that others solve their problems. Trust was considered a key barrier; many struggles to find someone to confide in, relying instead on acquaintances or strangers via social media for advice. Cultural norms around self-reliance, combined with discouraging messages from employment agencies (e.g., stating IMWs are unqualified for better jobs), reportedly undermine self-confidence. Consequently, IMWs rarely turn to their employers for support, fearing job loss. Additional barriers mentioned included shame, language difficulties and limited awareness of their rights. In some cases, IMWs reportedly return home when problems become too overwhelming.

Regarding mental health, stigma was considered a major barrier. Participants explained that mental health is rarely discussed with family or friends, particularly among men, who suppress their emotions to appear strong and to prioritise work. Participants also pointed out that among Polish IMWs, psychotherapy is often associated with being severely mentally ill. By contrast, turning to a priest was described as a familiar, culturally accepted way to cope with distress. Participants noted that if help is sought, it is typically first through such religious channels, and only once problems have escalated, a GP is contacted. According to participants, only in severe cases, a psychologist or psychiatrist is consulted.

Although participants described a range of everyday coping strategies, self-help tools, such as mental health apps, online resources or structured self-help programs, were rarely mentioned. When directly asked, most participants were unfamiliar with such resources or noted that they were not commonly used within their networks.

### Feedback on DWM/PM+ intervention

The intervention was discussed with both IMWs in the FGD (Round III) and professionals. Participants were unfamiliar with similar interventions for IMWs and reported several barriers when the DWM/PM+ intervention was explained. These barriers included language difficulties, digital illiteracy, reluctance to discuss problems and limited Wi-Fi access. While in-person sessions were preferred, travelling for PM+ sessions was considered another barrier. Although online delivery avoids this, participants worried about lower accountability and increased drop-out, believing IMWs are likely to sign up without following through. To improve engagement, participants emphasised the need for flexibility (e.g., evening and weekend sessions for shift workers), and privacy, particularly for those in shared housing.

Views on the intervention format were mixed, especially regarding the delivery format. Some appreciated DWM's low-threshold approach, but many were sceptical of psychological self-help, noting IMWs tend to prefer practical support. They doubted IMWs would engage with DWM exercises during the 15-min helper contact, suggesting a Q&A format instead. Participants preferred phone or text messaging over email. PM+ was valued for its human connection, with individual sessions favoured over group formats.

When discussing who should deliver the interventions, participants in the FGD debated whether psychology students or IMWs were most suitable. Students were seen as compassionate but inexperienced, while IMWs could relate but risked subjectivity. A combined approach was suggested: students guiding DWM, and IMWs with lived experience delivering PM+. Both FGD participants and professionals emphasised that helpers should be familiar with Dutch systems. Trust was seen as essential to intervention uptake: IMWs were thought to engage better when referred by familiar key figures. To ensure confidentiality, helpers should not come from the IMW's direct community, and emphasising the intervention's independence from employment agencies was considered crucial.

Participants also identified subgroups of IMWs who may need additional support. Temporary IMWs employed via agencies were considered particularly vulnerable yet hard to reach, as they prioritise day-to-day survival over long-term planning. Other groups included non-Dutch/English speakers remaining after their contracts ended, IMWs with children, retired IMWs, IMWs with physical issues, IMWs under 30, newer non-Polish IMWs (e.g., Moldovans with Romanian passports), and those supporting families back home, especially after job loss. Homeless IMWs were seen as urgently needing specialised care, especially during COVID-19.

### Adaptations

Based on study results, several adaptations were made to the DWM/PM+ stepped-care intervention (see Table 3). First, the target population was refined from all IMWs to Polish migrant workers, as they represented the largest group and each subgroup faces distinct challenges. Consequently, all intervention materials (i.e., for both participants and helpers) were textually and visually adapted to better reflect the lived experiences of Polish IMWs.

To address IMWs' transportation barriers and improve feasibility, the full intervention was delivered remotely. DWM is already an online intervention by design and remained web-based. PM+ was adapted for online (rather than in-person) delivery via Microsoft Teams, which is accessible via both laptop and smartphone. To enhance accessibility, PM+ sessions were shortened from 90 to 60 min and scheduled flexibly to fit participants' irregular work schedules. DWM support calls were also scheduled flexibly in agreement with the helper, and asynchronous messaging templates were added for DWM so that messaging could replace a support-call when needed. Remote delivery of PM+ required adjustments to the PM+ manual and additional helper training to address challenges related to digital engagement and ensuring quality support at a distance (e.g., Ensuring Quality in Psychological Support (Kohrt et al., 2020)). A challenge of online delivery was that participants needed to find a private space for sessions, which could be difficult in shared housing; the flexible scheduling aimed to accommodate this.

**Table 3.** Adaptations to the Doing What Matters in Times of Stress/Problem Management Plus intervention (DWM/PM+) stepped-care intervention, structured by the domains of the Bernal framework

| Domain | Operationalisation | Implementation | Rationale | Application | Intervention |
|---|---|---|---|---|---|
| Persons | Refine target population | - Narrowed from international migrant workers to Polish migrant workers. | - Enhance feasibility and recruitment by limiting to one language group. | Adaptation, implementation | DWM, PM+ |
| Persons | Refine helper and trainer/supervisor selection criteria | - Helpers: speak Polish, ≥12 yrs. of education, share cultural background and be empathic; - Trainers/supervisors: speak Polish. | - Enhance trust, communication and intervention delivery by matching culture and language. | Adaptation, implementation | DWM, PM+ |
| Language | Translation & terminology | - Translated all materials to Polish; - Replaced "client" with "participant". | - Ensure linguistic accessibility for participants, helpers and supervisors. | Adaptation, training, implementation | DWM, PM+ |
| Context | Address Pandemic Address socio-political factors | - No collaboration with employers; - Reinforced confidentiality throughout the intervention; - Added helper-guidance on supporting participants during the COVID–19 pandemic. | - Foster a safe therapeutic environment for sharing experiences; - Equip helpers to address pandemic-related stress. | Adaptation, training | DWM, PM+ |
| Content | Tailor content to participants' lived experiences and stressors | - Updated DWM app content and helper materials (scripts, examples, role-plays) to address relevant stressors; - Removed stressors irrelevant to the target population (e.g., community violence). | - Make intervention content meaningful and relatable; - Give helpers tools for context-specific problem support. | Adaptation, training, implementation | DWM, PM+ |
| Content | Ensure visuals are culturally appropriate | - Added illustrations representing the target population's context, including polish workers and European-style interiors. | - Ensure participants feel represented. | Adaptation | DWM, PM+ |
| Methods | Enhance helper guidance | - Added DWM app screenshots; - Included step-by-step instructions on navigating the DWM web app for interaction with the participant. | - Increase helper understanding of participant experience; - Provide clear support guidance. | Training | DWM |
| Methods | Integrate DWM strategies into PM+ | - Helpers ask about/apply DWM techniques in PM+. | - Strengthen continuity in stepped-care; - Reinforce DWM coping strategies. | Adaptation, implementation | PM+ |
| Methods | Shift intervention delivery mode | - Shifted intervention delivery method from face-to-face to remote. | - Enhance participation feasibility by removing travel barriers. | Adaptation, implementation | PM+ |
| Methods | Adjust session duration | - Shortened PM+ sessions from 90 to 60 min. | - Maintain engagement in remote format. | Adaptation, implementation | PM+ |
| Methods | Additional training for helpers | - EQUIP "Delivering Remote Services" training for PM+ helpers; - Added remote delivery guidelines to maintain professional and ethical standards (e.g., technology and privacy problems). | - Train helpers for remote delivery; - Maintain session quality and participant engagement. | Adaptation, training, implementation | PM+ |

Collaboration with employment agencies (e.g., for recruitment) was deliberately avoided to foster trust and minimise concerns about job security. Since this was the first time PM+ was delivered as part of a stepped-care intervention, helpers were also guided on integrating DWM techniques into PM+ to strengthen continuity between steps. No adaptations were made in the Bernal domains' Goals, Concepts, Metaphors or Language (beyond translation), as the intervention's overall aim and core psychological constructs needed to remain unchanged for trial standardisation and comparability across RESPOND sites. Deeper cultural adaptation of metaphors and language was not feasible due to the requirement for harmonised materials across other trials in the consortium.

## Discussion

This study explored the perspectives of IMWs and professional stakeholders on IMWs' problems, daily functioning and help-seeking behaviour, to inform the cultural adaptation of the DWM/PM+ stepped-care intervention in the Netherlands. Participants described employment insecurity, housing instability, administrative difficulties, barriers to (mental) healthcare and pandemic-related challenges. Many of these issues involved exploitative or illegal actions by employment agencies and persisted despite EU citizenship, compounded by limited legal awareness. Daily functioning centred on practical survival and mutual support, while help-seeking was hindered by stigma, job insecurity and low institutional trust. These insights informed adaptations to the intervention's content, delivery and training to improve its relevance, engagement and feasibility.

Our study revealed multiple interconnected dimensions of precarious employment among IMW's in the Netherlands, including dependence on employment agencies, employment insecurity, poor working conditions (e.g., long hours) and workplace discrimination (Ornek et al., 2022). These vulnerabilities were compounded by poor living conditions such as overcrowded housing and language barriers, reinforcing a sense of disempowerment. Although EU law guarantees equal treatment (European Commission, 2024), IMWs often do not exercise these laws because of their dependence on agencies, limited legal awareness and fear of losing work, leaving them structurally

vulnerable. Such conditions generate a cycle of disadvantage in which precarious employment, together with poor living conditions and migration-related stressors, undermine well-being and contributes to poor mental health among IMWs (Shirmohammadi et al., 2023). These dynamics align with broader patterns of labour exploitation, conceptualised as poor employment conditions, psychosocial hazards and disposability (Boufkhed et al., 2024), all clearly present in our data. Taken together, this illustrates how the intersection of migrant status and precarious employment makes IMWs especially vulnerable to exploitation and reduced well-being.

Our findings suggest that IMWs may delay seeking medical and formal psychological care due to financial constraints, language barriers, limited knowledge about the healthcare system and fear of job loss. These barriers echo previous research highlighting how long work hours, employer mediation, limited awareness of services and inadequate culturally adapted care restrict access to care (Thomson et al., 2015; Hennebry et al., 2016). Cross-national evidence resembles this pattern: Polish migrants in Norway reported difficulties with system navigation and distrust of GPs, while in the UK, healthcare choices were often based on familiarity, affordability and perceived quality, often resulting in migrants seeking care abroad (Czapka and Sagbakken, 2016; Troccoli et al., 2022). The transnational use of healthcare, also evident in our study, reflects a broader strategy to compensate for limited trust and accessibility in the host-country.

Together, these barriers highlight the need for accessible, low-intensity interventions that can be flexibly delivered in migrants' own language. DWM/PM+ appears well-suited to address IMWs' multi-dimensional problems, combining stress-management techniques with structured problem-solving strategies. However, substance use was frequently mentioned in our study and is not directly addressed. While PM+ may help manage contributing stressors, those with more severe addiction-related problems may require additional support. More generally, as a low-intensity intervention, PM+ relies on guidance and active engagement, which may be less suitable for IMWs experiencing serious mental health conditions requiring specialised care.

Study findings informed adaptations to DWM/PM+ for remote delivery, both to accommodate COVID-19 restrictions and improve accessibility for IMWs with irregular work schedules and limited transportation. DWM was converted into a guided e-health intervention and PM+ was offered remotely. Although guided digital interventions can be as effective as in-person care for treating depression (Moshe et al., 2021), participants noted challenges including digital illiteracy, privacy concerns, and dropout risk. Particularly vulnerable groups may experience lower motivation and trust in remote mental health services and prefer in-person support (Coomans et al., 2024), which could impact engagement and adherence. Moreover, effects of PM+ may be smaller in magnitude when delivered via video conferencing compared to in-person sessions (de Graaff et al., 2023). In response, we added asynchronous messaging templates for DWM, shortened PM+ sessions, provided extra training for helpers on delivering interventions digitally and reinforced confidentiality throughout the intervention, as assurance of privacy is a critical factor for trust in digital mental health care among migrant populations (Liem et al., 2021a).

To align with reported experiences and ensure feasibility, the intervention focused on Polish-speaking IMWs only. Although recruitment targeted IMWs, inclusion criteria did not specify job requirements, meaning all Polish-speaking individuals in the Netherlands in need of support could participate. While trial outcomes will clarify which subgroups can be reached and engaged, this broad approach limits the generalisability of findings to other IMW-groups. However, full contextual tailoring was limited as the intervention was designed for multiple RESPOND trials focusing on broader migrant populations in France and Italy (Melchior et al., 2023; Purgato et al., 2025).

This study has several strengths. To our knowledge, it is the first peer-reviewed qualitative study exploring the problems of IMWs in the Netherlands. Conducting interviews during COVID-19 provided valuable insight into pandemic-related problems. While timing may have shifted the focus from pre-existing issues, many identified problems were longstanding and exacerbated by the pandemic, revealing structural vulnerabilities that persist. Moreover, this pandemic-related focus was particularly relevant as the trial began during the pandemic.

However, our study also has some limitations. Given that most IMW participants were Polish, female, highly educated and had lived in the Netherlands for several years, the problem themes identified likely reflect the lived realities of this subgroup. Consequently, generalisability is limited, particularly to IMW groups beyond the Polish-speaking individuals represented in this study. For example, the temporary nature of seasonal work adds distinct layers of vulnerability, which persist beyond the COVID-19 pandemic (Augère-Granier, 2021; Bogoeski, 2022). Further differences exist between IMWs from various Eastern European countries, such as in wages (Felbo-Kolding and Leschke, 2023), contract type and return intentions (Snel et al., 2015). These factors may translate into different support needs or preferences for intervention delivery. Male perspectives were also underrepresented, as the FGD included only female IMWs. As the adaptations were informed by the perspectives and needs of our sample, these demographic and occupational characteristics may have influenced both the problems reported and the resulting adaptations. Nevertheless, the number and diversity of professionals interviewed likely helped to balance these gaps, since their focus was on IMWs with complex problems, which aligned with the intervention's target group.

Another limitation is that all data analysis and coding were conducted by a single researcher, and no independent verification (e.g., inter-coder reliability check) was performed. While we included both IMWs and professional stakeholders and used multiple interview formats, no formal data triangulation was conducted. The absence of these procedures may have increased the risk of researcher bias. Moreover, as the analysis was conducted by a researcher trained in clinical and academic models of mental health care, interpretations may reflect implicit biases associated with such frameworks. At the same time, training in global and cross-cultural mental health supported a broader and more contextually informed perspective during the analysis. Finally, while the DIME framework offered clear structure, it generated more detailed data than needed for the relatively modest intervention adaptations. Future studies may consider alternatives, such as the IDEAS (Integrate, Design, Assess, and Share) framework (Mummah et al., 2016), to better balance research depth with practical feasibility.

## Conclusion

This study highlights the structural vulnerabilities faced by IMWs, who often experience a high degree of dependency on employment agencies and interconnected problems beyond work-related stress. Language barriers were recurrent, underscoring the importance of psychological interventions in IMWs' native language. DWM/PM+ was culturally adapted for Polish migrant workers and redesigned

for remote delivery to improve accessibility and relevance. Future implementation of DWM/PM+ among other migrant populations or in new country contexts should involve renewed cultural adaptation, tailored to the specific linguistic, cultural and practical needs of each setting.

**Open peer review.** To view the open peer review materials for this article, please visit http://doi.org/10.1017/gmh.2025.10110.

**Supplementary material.** The supplementary material for this article can be found at http://doi.org/10.1017/gmh.2025.10110.

**Acknowledgements.** We would like to thank all participants whose input was invaluable in this study. Additionally, we thank Aleksander Kobylinski and Matylda Kępa for their efforts in the recruitment and conducting interviews with the participants, as well as Kevin Hamelink and Aleksandra M. Sobolewska for their help with transcribing and/or translating interviews.

**Author contribution.** R.R. was the primary researcher for this study and was responsible for data collection, analysis, and drafting of the manuscript. M.S. is the principal investigator of this study; she conceptualised the study, acquired funding, coordinates the RESPOND project (of which this study is a part) and supervised the study and writing process. A.W. also supervised the study and the writing of the manuscript. All authors critically reviewed and revised the manuscript and approved the final version.

**Financial support.** The RESPOND project has received funding from the European Union's Horizon 2020 research and innovation programme Societal Challenges under Grant Agreement number 101016127. The funder has no role in study design; collection, management, analysis and interpretation of data; writing of the report; and the decision to submit the report for publication. The content of this article reflects only the authors' views and the European Commission is not responsible for any use that may be made of the information it contains. Open access funding provided by Vrije Universiteit Amsterdam.

**Competing interests.** The authors declare no competing interests.

**Ethics approval.** This study has been reviewed and approved by the Scientific and Ethical Review Board (VCWE) at the Vrije Universiteit Amsterdam (VCWE-2021-006).

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
