## [Reviewer Report]

This manuscript provides an in-depth exploration of the mental health challenges faced by International Migrant Workers (IMWs) in the Netherlands and the strategies for adapting interventions. It holds significant academic value. The research methodology employs various qualitative data collection methods, incorporating the perspectives of both IMWs and relevant professionals, and utilizes cultural adaptation strategies to enhance the WHO’s mental health interventions (DWM and PM+). This makes a valuable contribution to the field of global mental health research. However, the article still requires major revisions in several areas, particularly regarding the study’s limitations, the specific effectiveness of the intervention adaptations, and the application of cultural adaptation theory.

1. Although the introduction mentions the challenges faced by IMWs, the summary of relevant literature is a bit lacking. References to existing research findings could be expanded and indicate what academic gaps this study fills. For example, literature on the mental health issues faced by immigrant workers and the available interventions could be added to further demonstrate the unique contributions of this study.

2. While the research design is well explained, there could be more detail on the data analysis process. For example, how was “saturation” determined? Were multiple researchers involved in data analysis? If so, was there inter-coder reliability testing?

3. Provide a more detailed discussion of the limitations regarding the choice of language groups (e.g., Polish migrant workers) and occupational groups (e.g., low-skilled workers), and how these factors might influence the study’s generalizability.

4. Providing more concrete examples or quotes from participants could strengthen the findings.

5. While the article addresses the cultural adaptation of the WHO interventions (DWM and PM+), the discussion of the adaptation of the intervention format is relatively underdeveloped. It would be valuable to further explore how the format and delivery of these interventions have been specifically adapted to meet the unique needs of International Migrant Workers (IMWs).For example, How have different formats (e.g., remote delivery, online platforms, face-to-face sessions) been adapted to address barriers such as language, time constraints, and privacy concerns, which are particularly important for IMWs? How were the intervention delivery methods tailored to better align with the cultural and practical realities of IMWs?

6. The conclusion could also include more on future research directions. For example, how can this intervention be adapted for different migrant populations, or how could it be implemented in other countries with similar challenges?

---

## [Reviewer Report]

Thank you for submitting this important and well-timed contribution. Your study addresses a significant and vulnerable population—international migrant workers—and provides rich qualitative data that contributes to the literature on culturally adapting psychological interventions for marginalised groups.

The rationale for the study is well articulated, particularly in linking the necessity of cultural adaptation to evidence-based psychological interventions like DWM/PM+. Your choice of the DIME model is justified, and your methods are, for the most part, clearly described. The thematic analysis is sufficiently detailed and aligns with your research aims. The contextual framing of COVID-19’s impact is also commendable, providing temporal relevance to your findings.

Nonetheless, there are areas that require revision to strengthen the manuscript:

First, the sample composition limits generalisability. Most participants were Polish, female, and relatively well-established in the Netherlands. While you acknowledge this in the Discussion, it deserves more prominent and critical treatment earlier in the paper. Consider explicitly discussing how this may have influenced the problem themes and intervention adaptation priorities.

Second, the methodological transparency could be improved. The absence of inter-coder reliability or triangulation in the data analysis should be acknowledged more openly as a limitation, particularly given the potential for researcher bias in inductive thematic approaches.

Third, the structure of the Results section, while thematically comprehensive, can become overwhelming. Some thematic overlaps (e.g., between housing, work, and COVID-19 restrictions) would benefit from clearer separation or signposting. Similarly, the Discussion section would be strengthened by greater synthesis across themes rather than treating them sequentially.

In terms of presentation, the abstract is dense and risks losing accessibility for non-specialist readers. Simplifying some language and focusing more explicitly on key takeaways—rather than repeating detailed methods—could enhance its impact. The Impact Statement is well-written and policy-relevant.

The tables, while rich, are excessively text-heavy and at times replicate narrative content from the main text. Consider summarising or streamlining them to focus on actionable findings and adaptations. Additionally, figure and table captions could be more informative to aid stand-alone interpretation.

Language and writing style are generally clear, though certain sections (especially the Discussion) would benefit from more concise expression and reduction of redundancy. Editing for stylistic consistency would also improve overall readability.

Please consider including references in the Introduction to support the need for cultural adaptation of psychological interventions in migrant populations:

Liu S, et al. Reliability and Validity of the Defeat Scale among Internal Migrant Workers in China: Decadence and Low Sense of Achievement. Healthcare (Basel). 2023 Mar 7;11(6):781. doi: 10.3390/healthcare11060781.

Giorgi G, et al. COVID-19-Related Mental Health Effects in the Workplace: A Narrative Review. Int J Environ Res Public Health. 2020 Oct 27;17(21):7857. doi: 10.3390/ijerph17217857.

Wang L, et al. Mental health and self-rated health status of internal migrant workers and the correlated factors analysis in Shanghai, China: a cross-sectional epidemiological study. Int Health. 2019 Oct 31;11(S1):S45-S54. doi: 10.1093/inthealth/ihz053.

Finally, your adaptation process—mapped against Bernal et al.’s framework—is a strong element of the study. However, several domains (e.g., “Goals,” “Metaphors”) are said to be unaltered without much justification. Briefly clarifying why changes in these domains were deemed unnecessary would add transparency to the adaptation logic.

In sum, this is a thoughtful and rigorous manuscript that makes a meaningful contribution to the field. With revisions aimed at improving clarity, methodological transparency, and structural flow, I believe it can be a valuable publication. I look forward to reviewing a revised version.

---

## [Reviewer Report]

1. The manuscript mention that data analysis was conducted by a single researcher. It might be helpful to include some mention of steps taken to ensure rigor to avoid potential bias.

2. In some places, the description of the intervention’s adaptation feels a bit scattered (e.g., changes to PM+ and DWM are discussed in separate sections). A more streamlined explanation of the adaptations could improve clarity for readers unfamiliar with the interventions.

---

## [Editor Report]

Dear Anke Witteveen,

Thank you for submitting your revised manuscript. We appreciate the contribution your work makes to the field.

After careful review, we are requesting a revision of your manuscript before it can be considered for publication. In particular, I would like to draw your attention to the points below:

Researcher Positionality: Since the coding and data analysis were conducted by a single researcher, please include a discussion of that researcher’s positionality. This context is important to help readers critically assess how the researcher’s background, assumptions, or experiences may have influenced the analytical process and interpretation of the data.

Data Triangulation: Please clarify whether any forms of data triangulation were used to strengthen the credibility of your findings. If triangulation was not used, we ask that you address this directly and discuss any implications for the trustworthiness of the analysis.

Stress-reduction strategies (p. 16): You note that participants identified activities such as sports, alcohol use, walking, meeting friends, and cooking as self-care strategies, while also stating that stress-reduction strategies (e.g., meditation) were reportedly absent. This raises a need for clarification as many of the listed self-care activities can reasonably be considered stress-reducing, even if they are not formalized or typically labeled as such. Please consider clarifying how you’re distinguishing between “self-care” and “stress-reduction strategies,” and whether this distinction is based on participants' own framing, theoretical definitions, or first author’s interpretation-- which can be further explained in the positionality. 

Help Seeking Behaviors (P.17): The authors note that “cultural associations of therapy with ‘being crazy’” and the role of priests as informal counselors often delay engagement with formal mental health services. While this is an important and culturally relevant insight, the current framing risks presenting these behaviors as deficiencies rather than as culturally grounded coping strategies.

Please consider clarifying the support typologies as it would be helpful to distinguish more explicitly between formal and informal types of emotional and psychological support. For example, consulting a priest or GP may not constitute formal mental health care, but these avenues can serve as important and culturally sanctioned sources of emotional support. Clarifying this typology will strengthen the analysis and avoid unintentionally devaluing these practices. It is also strongly suggested to address potential researcher bias as the current framing may reflect an implicit bias toward Western or clinical models of mental health care. We suggest acknowledging this perspective explicitly and either addressing the bias or incorporating literature that validates informal, community-based, or culturally specific sources of emotional support (e.g., pastoral counseling, family networks, peer support within faith communities).

The claim that IMWs are often unaware of self-help tools would benefit from clarification or citation. If this is a key point, please consider whether participants explicitly indicated this, or whether it is an interpretation that requires further justification-- it seems as though you describe several tools used by this sample. 

Coding and Analyses: Given that coding and analysis were conducted by a single researcher, we strongly recommend involving an additional reviewer or auditor in the data analysis process. This would improve the interpretations and allow for a more nuanced discussion of the findings, particularly when dealing with culturally sensitive topics.

These revisions align with Reviewer 1’s comments and are necessary to enhance the transparency and rigor of the manuscript.

Please include a response letter detailing how each point (including other reviewer comments) has been addressed in your revised manuscript.

We look forward to receiving your revised submission.

Sara Romero

---

## [Reviewer Report]

1. In lines 429-440 and lines 449-457, you discuss the adaptation of the intervention, particularly the shift to remote delivery to address barriers such as transportation and scheduling. However, I suggest further exploration of the challenges that may arise from implementing remote interventions. While remote delivery offers flexibility, some migrant workers might prefer face-to-face support, especially when it comes to building trust and fostering engagement. It would be valuable to consider whether these adaptations could impact the intervention’s overall effectiveness, particularly in terms of participant engagement and potential risks of lower adherence to the program.

2. Regarding the representativeness of the sample, while you acknowledge that the sample was primarily Polish-speaking and female, I recommend further discussion of how other groups such as male migrant workers or those from different countries might face distinct challenges. Expanding on how these different demographic factors could influence both the problems faced by IMWs and the intervention’s adaptation would strengthen the applicability and generalizability of your findings